# The Comparison of the Effects between Continuous and Intermittent Energy Restriction in Short-Term Bodyweight Loss for Sedentary Population: A Randomized, Double-Blind, Controlled Trial

**DOI:** 10.3390/ijerph182111645

**Published:** 2021-11-05

**Authors:** Manwen Xu, Ji Li, Yan Zou, Yining Xu

**Affiliations:** 1Department of Integrated Traditional Chinese and Western Medicine, Huashan Hospital, Fudan University, Shanghai 200040, China; xumanwen@huashan.org.cn (M.X.); liji@huashan.org.cn (J.L.); zouyan@huashan.org.cn (Y.Z.); 2Faculty of Sports Science, Ningbo University, Ningbo 315211, China

**Keywords:** continuous energy restriction, intermittent energy restriction, bodyweight loss plan, diet intervention, RCT

## Abstract

Objective: To compare the effects of continuous energy restriction (CER) and intermittent energy restriction (IER) in bodyweight loss plan in sedentary individuals with normal bodyweight and explore the influence factors of effect and individual retention. Methods: 26 participants were recruited in this randomized controlled and double-blinded trial and allocated to CER and IER groups. Bodyweight (BW), body mass index (BMI), and resting metabolic rate (RMR) would be collected before and after a 4-week (28 days) plan which included energy restriction (CER or IER) and moderate-intensity exercise. Daily intake of three major nutrients (protein, carbohydrate, fat) and calories were recorded. Results: A significant decrease in BW and BMI were reported within each group. No statistically significant difference in the change of RMR in CERG. No statistically significant difference was reported in the effect between groups, neither as well the intake of total calories, three major nutrients, and individual plan retention. The influence factors of IER and CER are different. Conclusion: Both CER and IER are effective and safe energy restriction strategies in the short term. Daily energy intake and physical exercise are important to both IER and CER.

## 1. Introduction

Energy restriction is necessary for bodyweight loss plans for athletes who focus on optimizing their body composition such as physique competitors or Olympic weightlifters. To achieve low levels of body fat and keep as much fat-free mass as possible, athletes typically follow 8 to more than 20 weeks of daily diets with energy deficiency in which energy expenditure is increased and caloric expenditure is decreased [1,2,3,4,5,6,7,8,9,10]. Many different strategies could be used to make daily energy deficiency, among these strategies, continuous energy restriction (CER) and intermittent energy restriction (IER) are wildly used as pre-contest strategies in physique competitors and Olympic weightlifters. CER requires reducing a daily energy intake relative to bodyweight maintenance requirements; alternatively, IER uses alternating periods of energy restriction with periods of greater energy intake that are sometimes referred to as “refeed” periods or “cheat days” within the fat-loss plan [11,12].

Previous studies have shown that the effects of CER and IER in bodyweight loss plans are similar for professional athletes and sedentary individuals but slightly different in the improvement of body composition. A study made by Campbell’s team and published in abstract form investigated the effects of a 2-day refeed in dieting 14 resistance-trained males and 13 resistance-trained females seeking to optimize their physiques [13]. In this study, there was a 7-week diet phase in which one group was randomly assigned to reduce their caloric intake by 25% per week for 7 consecutive weeks (CERG). By contrast, the other group (IERG) reduced their caloric intake by 35% for 5 of 7 days per week while including a 2-day increase in caloric intake (in the form of carbohydrates only) for 2 consecutive days per week. At the end of the week, both IERG and CERG reached an energy restriction at 25%, and at the end of the 7-week dieting intervention, both groups had significant reductions in fat mass and bodyweight, but the IERG retained more fat-free mass compared with the CERG by 0.9 kg. Furthermore, the CERG experienced a significant decrease (4%) in resting metabolic rate (RMR) as compared to baseline values, whereas the IERG maintained their resting metabolic rate during the diet intervention.

However, some other previous studies also reported that CER would induce a down-regulation of total daily energy expenditure, which includes resting metabolic rate (RMR, 60–70%), non-exercise activity thermogenesis (NEAT), thermic effect of food/diet-induced thermogenesis (TEF/DIT, 10–15%), and exercise activity thermogenesis (EAT, 5–15%). Each of these components has been shown to decrease in response to body-weight loss and/or energy restriction. For example, it has been reported that EAT and NEAT would be suppressed as a response to bodyweight decrease in obese individuals [14,15,16,17]. The relative magnitude of the TEF would not change with energy restriction, but the overall reduced energy intake would still decrease the absolute magnitude of the TEF [18,19]. Among the variables comprising total daily energy expenditure, RMR has been the most studied in the physique competitors. In these published case studies where RMR of physique competitors have been tracked during contest preparation, there has been a suppression of the RMR with an average reduction of approximately 18% (ranging from 9% to 47%) congruent with fat loss and energy restriction [1,4,6,20,21].

Meanwhile, studies also reported CER might induce some adverse physiological effects. For example, a discussion of the impaired physiological functioning that manifests itself was put forth by the International Olympic Committee (IOC) in the form of a consensus statement and is referred to as a syndrome titled “Relative Energy Deficiency in Sport (RED-S)”. RED-S means a relative energy deficiency and includes impairments of RMR, menstrual function, bone health, immunity, protein synthesis, and cardiovascular health [22]. In other words, low-energy availability is the likely factor responsible for the impairments observed in RED-S. Considering the preparation of physique athletes and Olympic weightlifters would undertake a combination of energy restriction and a concomitant increase in exercise volume load, the likelihood of this type of athlete experiencing symptoms of RED-S would elevate.

One of the potential benefits of incorporating an IER dietary strategy is to offset some of the adverse physiological effects that CER can exert. Although research investigating IER in physique athletes is in its infancy, there are some data and optimism to suggest that such strategies could prevent RED-S [2,3,18]. Chappell’s team reported that 10 out of 32 male competitors and 8 out of 16 female competitors consumed periodic “cheat meals” during their contest preparation, furthermore, one of the 32 males and 4 of the 16 females used refeed strategies during their contest preparation [3]. Refeed strategies were also reported by Mitchell’s team among 4 of 9 bodybuilders as part of their contest preparation [2].

To sum up, the bodyweight loss and energy restriction practices that athletes themselves to for competitive performance elicit physiological homeostatic responses. These responses include metabolic adaptations that are manifested through the suppression of metabolic rate. Nevertheless, for elite athletes, unfavorable responses to fat loss and energy restriction would be reversible with post-competition weight gain and increased energy intakes. Therefore, for athletes and sedentary individuals, both IER and CER would be an effective strategy for a bodyweight loss plan.

Studies also have compared the effects of IER and CER on obese or overweight individuals. It has been suggested that active lean individuals may benefit more from IER strategies due to their different metabolic statuses as compared to the overweight or obese sedentary populations [11]. Besides, although the practice of IER has been around for many years in the physique world, it is not until recently that research on these practices has been performed with these populations [23,24].

The COVID-19 pandemic has forced public health authorities to impose lockdown measures as an epidemiological containment strategy, causing people to lack daily physical activities and a rapid increase in the number of sedentary individuals worldwide [25]. The sedentary individuals are not overweight or obese and do not have a professional sports background, but they have a common desire to lose some bodyweight quickly after being inspired by global public health campaigns. A common feature of their desire was to lose a small amount of bodyweight in a short period to make them feel better about themselves. In other words, for the sedentary population with normal bodyweight, their psychosocial needs outweigh their clinical needs. 

On one hand, studies have shown that even for individuals whose BMI is in the normal range, a tiny decrease in BMI could still reduce the risk of chronic disease and all-cause mortality significantly. Besides, in the social environment where body-shaming is prevalent, maintaining a high degree of self-satisfaction with body shape under the premise of health could increase personal wellbeing and improve the quality of life. It means that short-term bodyweight loss, even for those who are clinically “normal”, is of great value and importance [26,27,28]. On the other hand, although clinically speaking, individuals with normal BMI, especially the young, do not need to lose bodyweight in a short term, it is still very common to lose bodyweight in a short term from the perspective of psychosocial needs and daily practical application. For example, to attend a wedding, a dinner party, an important interview, etc., in a few weeks, people often encounter similar scenarios and hope to reach their target bodyweight as soon as possible, to look better and feel better about themselves [29,30,31].

Unfortunately, it is unclear whether the IER and CER were also effective in sedentary individuals with normal bodyweight. There are very few studies on bodyweight loss plans in this population. To fulfill the academic gap, this study is conducted to compare the effects of CER and IER in bodyweight loss plans in sedentary individuals with normal bodyweight and explore the influence factors of effect and individual retention. Since there is some initial evidence indicating that IER may help to attenuate a decrease in fat-free mass and RMR that would be beneficial for sedentary individuals, it could be hypothesized that the same phenomenon will occur in a sedentary individual with a normal bodyweight.

## 2. Method

### 2.1. Participants Recruitment and Ethics

From 31 May 2021 to 27 June 2021 (4 weeks), the first researcher published the recruitment information of volunteers online, and the volunteers would register their names and contact information. Necessary personal information such as name, age, gender, and daily physical activities was collected from the volunteers between 28 June and 4 July (one week) and was screened according to inclusion and exclusion criteria for the trial by a second researcher. In this period, all volunteers would not know anything about the trial. They would just know there was a bodyweight loss plan to follow. Volunteers who met the inclusion criteria (participants) were randomly allocated by the second research group from 5 July to 11 July (1 week). In this period, all participants still didn’t know anything about the trial.

#### 2.1.1. Inclusion Criteria

The inclusion criteria of participants were as follows: (1) From 18 to 60 years old; (2) free from endocrine, metabolic, neuromuscular, and musculoskeletal disorders; (3) BMI from 19 to 28 [32]; (4) without any diseases that are not clinically recommended for physical activity; (5) not engaged in any physical activity with moderate or above intensity for at least 6 months.

#### 2.1.2. Exclusion Criteria

The exclusion criteria of participants were as follows: (1) under 18 years old or over 60 years old; (2) with endocrine, metabolic, neuromuscular, or musculoskeletal disorders; (3) be clinically required not to participate in any physical exercise; (4) be asked to participate in the trial involuntarily; (5) participated in physical activity with moderate or above intensity within the last 6 months.

#### 2.1.3. Ethics Approvement

All participants had written informed consent and the study was approved by the Institutional Ethics Committee of Ningbo University (NO.20210601).

### 2.2. Protocol

#### 2.2.1. Randomization and Blinding

The randomization was made by the Random Number Generators Function of the SPSS Software (Version 23.0, SPSS, Inc., Chicago, IL, USA) and the trial was registered in the Trial Registry of Ningbo University, the registration number was NBURT.20210601. All participants would be randomly allocated into continuous energy restriction group (CERG) and intermittent energy restriction group (IERG) by the second researcher. All participants would not know which group they were allocated to and only know they would have a bodyweight loss plan to follow in the next several weeks. The third and fourth researchers would be randomly allocated into CERG and IERG by the same randomization method to make the trial double-blind. The researchers in IERG and CERG would introduce the procedures and rules of the trial, teaching the participants to record and report their daily diet and physical exercise. 

The first researcher, who recruited and screened the participants, would not know the allocation of each participant and the third and fourth researchers. The third and fourth researchers also didn’t know anything about the other experimental group. Only the second researcher knew the allocation of each participant.

#### 2.2.2. Intervention(s)

From 12 July to 18 July (one week), participants learned the trial process and how to report their daily diets and physical exercise. The third and fourth researchers should provide a guide to make ensure that each participant could be familiar with the trial’s process and learn to self-report before the formal trial began. The third and fourth researchers should educate all the participants one by one, collect their self-report every day, dispose of daily data, and give the daily data to the first researcher after concealing the personal information of each participant. From 12 July to 8 August (4 weeks), the formal trial was conducted for 4 weeks.

Intermittent Energy Restriction (IER)

The IER strategy would be conducted in the form of “Diet Refeeds”. Traditionally, the Diet Refeeds was an energy restriction strategy with a daily energy intake that reaches a specific target of calories and nutrients below their estimated weight maintenance energy requirements for 1–3 days [2], and then have a cheat day, which is an entire day of eating without regard to quantity or nutrient composition. However, according to the desire of the participants, which was to lose bodyweight as quickly as possible, the Diet Refeeds strategy used in this trial would let the participants in IERG intake 800 kcal in each energy-restricted day, which was an extremely low daily energy intake, with an error tolerance at 40 kcal. The intake may be measured at the end of each day, there are no predetermined goals to achieve in terms of nutrient intake. Cheat days might include a string of cheat meals and most likely result in a significantly higher than normal consumption of calories that usually come from carbohydrates and fats. In IERG, each day with an energy intake from 760 kcal to 840 kcal would be regarded as a “valid day”, as well as each cheat day with a report of energy intake and physical exercise. The progress is provided in Figure 1.

Continuous Energy Restriction (CER)

The CER strategy would be conducted in the form of a “diet break”. Diet breaks—A diet strategy where a continuous string of 4+days to several weeks of weight maintenance calories (or slightly above) are consumed as part of the fat loss plan with a specific target of calories and macro/micronutrients for each day. On each day, the participants in CERG were asked to have a daily energy intake with a deficiency (exclude EAT) of 500 kcal. After allocation, the researcher would calculate the range of daily energy intake of each participant to reach 500 kcal daily energy deficiency (exclude EAT) with a tolerance of error at 5% (450–550 kcal). 

In CERG, the day would be considered as a “valid day” if the participant’s daily energy intake reaches 500 kcal daily energy deficiency (exclude EAT) with a tolerance of error at 5% (450–550 kcal) and be considered as “invalid day” if it was not in this range.

Schofield’s Equation, which is recommended by the WHO as the best formula for the Asian population with high reliability and validity, was used to calculate the RMR in baseline and endpoint. According to the consensus statement of the National Health Commission of the People’s Republic of China, when it comes to the Chinese population, the result of Schofield’s Equation should be multiplied by 95% [33,34,35,36]. The percentage of the sum of RMR and TEF/DIT to the total daily energy expenditure would be set at 80% (RMR = 70% and TEF/DIT = 10%) when calculating the daily energy deficiency. The BMI, RMR, and Target Daily Energy Intake (TDEI) would be calculated by the formula as follows:BMI (kg/m^2^) = Bodyweight/Height^2^(1)
RMR (kcal) = [63 × Bodyweight (kg) + 2896]/4.1828 × 95%, Male, age 18–30(2)
RMR (kcal) = [48 × Bodyweight (kg) + 3653]/4.1828 × 95%, Male, age 31–60(3)
RMR (kcal) = [62 × Bodyweight (kg) + 2036]/4.1828 × 95%, Female, age 18–30(4)
RMR (kcal) = [34 × Bodyweight (kg) + 3538]/4.1828 × 95%, Female, age 31–60(5)
TDEI (kcal) = (RMR/0.8) − 500(6)

Physical Exercise

According to the consensus statement of The American College of Sports Medicine (ACSM), the participants would also be suggested to have physical exercise with a total volume load that reaches a metabolic energy expenditure from 2400 kcal to 2600 kcal every week [37]. There were no requirements for the type or duration of daily physical exercise, but participants were advised to try to choose only one type of physical exercise at a time.

### 2.3. Data Collection

From 9 August to 15 August, the first researcher disposed, calculated, and analyzed all the data and give personal results to the second researcher. The second researcher would make detailed self-recorded report summaries to all participants who completed the trial. The template of detailed self-recorded report summaries would be provided in the Appendix A.

#### 2.3.1. Participants Characteristics

During the participants’ recruitment and screening, the individual’s information of gender, age, body height, and bodyweight would be collected and recorded.

#### 2.3.2. Daily Data Collection

Participants were asked to report their daily dietary intake, including the type and amount of food and drink they consumed. Participants were also asked to report the type and duration of their physical exercise every day. The researchers would calculate participants’ energy balance condition for the day based on their self-report, the nutrient content and calorie density of their food and drink, and the rate at which calories were consumed for different types of physical exercise. The nutritional composition and calorie density of food or drink and the calorie consumption rate of different kinds of physical exercise in the calculation process were referenced from the National Health Commission of the People’s Republic of China database. At the same time, the cumulative weekly energy expenditure of physical exercise was calculated and reported back to the participants by the third and fourth researchers without any guidance, supervision, or encouragement.

#### 2.3.3. Participants Characteristics

The name, gender, age, bodyweight, and height would be recorded, and the BMI and the RMR would be calculated and recorded at baseline. The bodyweight would be recorded again at the endpoint. The bodyweight at baseline will be based on the fasting bodyweight of the participants on the morning of the first day in the plan. BMI and RMR will be calculated and recorded by using the formula provided above according to the participant’s bodyweight and height.

#### 2.3.4. Individual Plan Retention

After the whole trial, participants would be regarded as lost to follow-up whose sum of invalid days was greater than 14. Daily intakes of the three major nutrients, energy, exercise energy expenditure, and energy deficiency was recorded. At the end of the trial, the total and average of the three major nutrient intakes, energy intakes, exercise expenditure, and energy deficiency would be calculated. The average value was calculated according to the total value divided by the sum of valid days, which would be used to assess the individual retention of the plan.

### 2.4. Statistical Calculation and Analysis

The daily energy intake (DEI), the daily energy deficiency (DED), and relevant total-value (energy intake, TEI; total exercise activity thermogenesis, TEAT; total energy deficiency, TED) would be calculated by the formula as follows:DEI (kcal) = 4 × Protein (g) + 4 × Carbondydrate (g) + 9 × Fat (g)(7)
DED (kcal) = (RMR/0.8) − DEI + EAT(8)
TEI/TED/TEAT (kcal) = DEI/DED/DEAT × Sum of Valid day(9)

The daily data of each participant would be calculated by the third and fourth researchers in IERG and CERG. 

Changes in bodyweight, BMI, RMR, EAT, energy intake, energy deficiency, and individual retention of the plan within each group would be statistically analyzed by using the paired *t* test, whereas the difference in baseline characteristics and difference in changes in bodyweight, BMI, RMR, EAT, energy intake, energy deficiency, and individual retention of the plan between groups or subgroups were statistically analyzed by using the Student’s *t* test. Variance homogeneity would be tested by Levene’s test and the statistical power of the Student’s *t* test would be calculated to check the probability of the Type II Error.

The Pearson distance correlation coefficient would be calculated within the protein, carbohydrate, fat, EAT, energy intake, and energy deficiency both daily and a total of participants in each group. Moreover, the Pearson distance correlation coefficient would be calculated within all data collection of all participants (combination of IERG and CERG). In the Pearson distance correlation, two variables with a correlation coefficient from 0.5 to 0.8 would be considered moderately correlated, and two variables with a correlation coefficient greater than 0.8 would be considered highly correlated. The regression functions would be made based on the result of Pearson distance correlation coefficients. The regression function of daily energy deficiency in each experimental group would be constructed by daily intake of energy, nutrition, or EAT, whereas that of the BMI percentage change would be constructed by the baseline information and the intake of energy, nutrition, and EAT in the whole plan.

The statistical analysis would be made by the first researcher who didn’t know the allocation of participants. The SPSS Software 17.0 (SPSS, Inc., Chicago, IL, USA) was used for all analyses. Data would be presented as means and standard deviations except if otherwise specified and considered statistically significant at *p* < 0.05.

## 3. Results

### 3.1. Participants

After recruitment and screening, 26 participants were included in this trial (IERG n = 12, CERG n = 14) and 18 participants completed the whole plan (IERG n = 8, CERG n = 10). The progress of the flow diagram of the whole trial is provided in Figure 2. The participants’ characteristics of each experimental group were provided in Table 1. According to the result of the Student’s *t* test, there is a statistically significant in age and body height between IERG and CERG. Nevertheless, there is no statistically significant difference in the baseline bodyweight, BMI, and RMR between IERG and CERG, meaning that the two groups have similar bodyweight conditions.

### 3.2. Effect on Bodyweight Loss Plan

The effect of bodyweight management plans would be evaluated by the results of the paired *t* test within each group is provided in Table 2 and Figure 3. The details of energy balance and the result of the Student’s *t* test between groups with its statistical power (1-β) are provided in Table 3. 

According to the results, the bodyweight and BMI have decreased and reached a statistical significance both in IERG, CERG, and total participants (*p* < 0.05). However, there is no statistically significant difference between IERG and CERG (*p* = 0.859 and *p* = 0.647). The RMR had a statistically significant decrease in IERG (*p* < 0.05) but not in CERG (*p* = 0.128). However, the change of RMR between IERG and CERG didn’t reach a statistically significant (*p* = 0.582).

All the *t* test results of the difference in bodyweight, BMI, and RMR showed a good statistical power (greater than 0.80). However, the *t* test results of the TEI, TEAT, and TED showed a low statistical power (less than 0.80).

### 3.3. Individual Plan Retention

The individual plan retention, which is represented by the sum of valid days in the whole plan, would be evaluated by the result of the Student’s *t* test between groups with its statistical power that provided (1-β) in Table 4. According to the result, there is no statistically significant difference in the sum of valid days between IERG and CERG (*p* = 0.122). The *t* test results of the TEI, TEAT, and TED showed a low statistical power (less than 0.80). 

### 3.4. Influence Factors Analysis

The exploration of the influence factors of effect and individual retention would be made by the analysis of correlation within variables and regression functions. The results of the Pearson distance correlation coefficient analysis would be provided in Table 5 and the result of the comparison between the mean value of daily statistics between valid and invalid days would be assessed by Student’s *t* test and provided in Table 6. According to what was mentioned in the method section, the regression functions would be made one by one based on the variables that have a moderate or high correlation with the BMI change and percentage change of the baseline value in each group and that in total participants. 

The result of the correlation analysis between the effect of the plan and the variables in the daily report was provided in Table 7 in the form of a league table. In the league table, each result represented the correlation coefficient between the column-defining effect and the row-defining variable.

The results of regression analysis are provided in Table 8. According to the results, the regression functions in total and subtotal participants in this trial are as follows:PCBMI (% BBMI) = 10.319 − 0.172 × DF, IER(10)
PCBMI (% BBMI) = 3.809 − 0.244 × Age − 0.722 × BBMI + 0.015 × BRMR + 0.032 × TED − 0.001 × DED, CER(11)
PCBMI (% BBMI) = 9.660 − 0.152 × DF, ER(12)
DED (kcal) = 1824.660 − 0.982 × DEI + 1.239 × EAT, IER(13)
DED (kcal) = 648.899 − 1.033 × EAT, CER(14)
DED (kcal) = 1649.280 − 0.794 × DEI + 1.232 × EAT, ER(15)

## 4. Discussion

The objective of this study is to compare the effects of CER and IER in bodyweight loss plans in sedentary individuals with normal bodyweight and explore the influence factors of effect and individual retention. The main finding of this study is that both CER and IER are similarly effective short-term bodyweight loss plans without a significant difference in individual retention. However, the IER might induce a decrease in RMR whereas the CER wouldn’t. Moreover, different energy restriction strategies might require different focuses to make enough energy deficiency and fit for a different population. 

In terms of bodyweight-related parameters, both CER and IER could significantly reduce BW and BMI, and there is no significant difference in short-term effects between them. This phenomenon is consistent with the results of studies in athletes and the sedentary population [38]. For example, in 1981, Shubin’s team claimed that the regulation of the wrestlers’ bodyweight through a hypocaloric diet would favor the working capacity and functional state of the athletes [39]. At the same time, since BMI is an effective indicator of the risk of obesity and overweight [40,41], the results of this study indicate that both CER and IER could effectively reduce the risk of obesity or overweight. 

On the other hand, there is a statistically significant difference in TEI and TEAT between groups. This result might indicate that the similarity in the effect on bodyweight loss of the two strategies is due to the same TED they made but their pathway to create energy deficiency might be different. According to the result, the IER strategy inclines to intake less energy and have a low or moderate level of exercise activity, whereas the CER strategy inclines to create its energy deficiency through a high level of physical activity.

When it comes to metabolic adaption, there is no statistically significant difference between IERG and CERG in the change of RMR, however, the average RMR in participants of IERG has been decreased significantly. Due to the equation for calculating RMR in this study being based on the body composition characteristics of a large sample population and the BMI of the participants in this study was within the normal range, it could be assumed that the body composition of the participants was consistent with the body composition characteristics of the population applicable to this equation. Considering that the decrease of bodyweight in IERG and CERG were similar, it could be inferred that the difference in RMR changes between IERG and CERG might be induced by just mathematical factors and could not represent metabolic adaptions. Previous studies suggested that, compared to the overweight or obese sedentary populations, active lean individuals might benefit more from IER strategies due to their different metabolic statuses [11]. At the same time, the recently proposed “constrained energy expenditure model” and the experimental basis of this model in humans included cross-sectional data might provide alternative explanations, which is that the daily energy expenditure would be regulated and that free-living daily energy expenditure adjusted for body composition is relatively constant [42,43,44]. Similarly, it also could not be proved that there is no difference in metabolic adaption between IERG and CERG. First, a 4-week CER plan may not be long enough to make a significant difference in RMR, the previous studies, which reported metabolic adaption in athletes or sedentary population, usually have a longer bodyweight loss plan to reach the target bodyweight. Therefore, athletes and sedentary individuals would use the week as the unit of their Diet Refeeds cycles. For example, they would set a break week after 4 weeks of energy restriction. It means that the duration for the athletes or sedentary individuals to lose bodyweight would be much longer than that in this study [45,46,47]. Second, because athletes or sedentary individuals are often exposed to a high-volume load of training or physical exercise, they need to intake more energy in their IER and have a more rigid limitation in the nutrient intake than the participants in this study. For example, for athletes, their calorie intake is often equal to or slightly higher than their bodyweight maintaining value, whereas the participants in this study were intaking an extremely low calorie in their energy restriction days. Finally, the proportions of the three main nutrients of athletes, bodybuilders, or physique competitors might differ from those of the participants in this study. In this study, the researchers did not have specific requirements for the intake of the three main nutrients of the participants, while athletes, bodybuilders, or physique competitors generally have a rigid and meticulous dietary program [48,49]. Future studies should explore what is the best energy restriction strategy for a rapid bodyweight loss plan in sedentary people with normal bodyweight by tracking the change of their body composition and metabolic hormone response.

To individual plan retention, the rates of loss to follow-up are 33.3% in the IERG and 28.6% in the CERG. There is no significant difference in the average of the total days with energy restriction between the two groups. It indicates that a 4-week IER and CER strategies might have no difference in individual retention, and both could achieve a good completion. Future studies should explore the changes in completion of other energy restriction strategies with different duration and the differences in completion difficulty among different energy restriction strategies. Participants in IERG and CERG whose completion over 50% had significant differences in average total carbohydrate intake, total protein intake, total fat intake, total energy intake, and total EAT, but there was no significant difference in the average of total energy deficiency between the two groups. It shows that the total energy deficiency that IER and CER strategies created would be similar under similar completion conditions. This may partly explain why there was no significant difference in average bodyweight decrease between the two groups. However, the average of total energy intake and total EAT of IERG participants are significantly lower than those of CERG participants, indicating that although there is no significant difference between the total energy deficiency of IERG and CERG, IER and CER strategies may have different ways to create energy deficiency. A subgroup analysis of the sum of valid days (VD) and invalid days (IVD) in each IERG participant found significant differences in the average of daily EAT between VD and IVD in IERG, but not in daily energy intake and daily energy deficiency, which may mean that in the short-term, an approach to extremely low-calorie diets may not affect the conduction of a moderate-intensity exercise with 30 to 60 min every day is sedentary individuals with normal bodyweight. Therefore, it could be concluded that a short-term Diet Refeeds strategy with extremely low caloric intake could combine with 30 to 60-min of daily physical activity under a moderate intensity, which is recommended by the ACSM [37,50]. However, this conclusion has some limitations. First, self-reports tend to underestimate the intake [51], and second, there is insufficient evidence to prove the safety of the strategy since the researchers in this study did not measure relevant biochemical and physiological indicators [6,52,53,54]. During the study, several participants reported minor adverse effects such as constipation, fatigue, and mild anxiety, so caution should be exercised in the application of the IER strategy with an extremely low caloric intake. However, the *t* test results of the TEI, TEAT, and TED showed a low statistical power (0.67), suggesting that the result without statistically significant difference in the sum of valid days between groups might be due to the small sample size and the short duration of the trial.

Subgroup analysis of days with physical exercise (ED) and days without physical exercise (NED) of IERG (according to whether the daily EAT is greater than 100 kcal or not) found that there was a significant difference in daily energy deficiency between ED and NED, but no significant difference in daily energy intake and nutrients intake condition. It might indicate that the exercise intervention used in this study may not affect the condition of energy and nutrient intake of the IER strategy. It is widely feared that physical exercise under energy restriction would cause changes in appetite, which will affect the effect of dietary intervention and health [55,56,57,58]. However, in this study, physical exercise did not affect the intake of energy and main nutrients. Therefore, it could be inferred that an energy restriction strategy with the maintenance of moderate-intensity physical exercise in the short term would not affect the diet intervention. There are also limits to this conjecture. First, the researchers did not analyze hormonal changes [1,18,59], and second, self-reported errors also could not be ignored. Therefore, caution should be exercised in practice. In the future, energy intake and physical exercise of long-term IER strategy should be measured, and changes in relevant physiological and biochemical indicators of participants should be monitored.

The study also found that in IERG, the average daily EAT in days with energy restriction reached 383.4 kcal, while the average difference of daily energy deficiency was 448.6 kcal. At the same time, in IERG, the correlation between daily energy intake and daily EAT was low (0.149), but the correlation between daily energy intake and daily energy deficiency was moderately negative (−0.677). Meanwhile, the daily energy deficiency also has a moderate positive correlation with daily EAT (0.506). It seems to indicate that, under the IER strategy, the daily energy deficiency depends on the value of daily energy intake and daily EAT, which is reasonable and supported by a large amount of evidence [18,60,61]. For the whole plan, there is a moderate positive correlation between the decrease of BMI and average of individual plan retention (0.514), a moderate negative correlation with total fat intake (−0.574), a high negative correlation with daily fat intake (−0.723), and a moderate negative correlation with daily energy intake (−0.646). There was not a high correlation with age, body height, and other baseline data, with average daily and total carbohydrate and protein intake, neither with daily EAT and total EAT. The results were the same when the decrease in BMI was converted into a percentage decrease. Considering the moderate correlation between individual plan retention and total energy deficiency and daily energy intake (0.569 and −0.611), and the high correlation between individual plan retention with total energy deficiency and total EAT (0.725 and 0.624), the small negative correlation between individual plan retention and daily energy intake (−0.290), the moderate correlation between daily energy intake and daily carbohydrate intake, and the high correlation between daily energy intake and daily fat intake (0.561 and 0.822), it could indicate that the main factor affecting the effect of IER strategy is the condition of daily execution, because a good execution could ensure an extremely low daily energy intake and the total energy deficiency in the end. Moreover. the key to ensuring extremely low daily energy intake might be to control the intake of carbohydrates and fat. At the same time, to achieve better bodyweight management effect of IER strategy, daily EAT should also be guaranteed, and daily fat intake should be controlled to achieve enough daily energy deficiency, which is consistent with the results of previous studies [62,63,64].

In CERG, the average difference of daily EAT between ED and NED was 477.4 kcal, while that of the average daily energy deficiency between ED and NED was 544.7 kcal. By analyzing the correlation between daily data of participants in CERG, it is found that in CERG, the correlation between daily energy intake and daily EAT is relatively low (0.216) and that between daily EAT and daily energy intake is also relatively low (−0.357), while the correlation between daily EAT and daily energy deficiency is much higher (0.759). According to this result, it could be inferred that in CERG, daily energy deficiency might depend more on the daily EAT. Considering that in CERG, the part of daily energy deficiency excluding daily EAT was pre-set at 500 kcal, so the result is consistent with the expectation of the study. 

The results of overall data correlation analysis in CERG were significantly different from that in IERG. In CERG, there was a low correlation between the average BMI decrease and individual plan retention (−0.052), as well as a low correlation with total nutrients intake and total EAT. However, in the CERG, the average BMI decrease had a moderate correlation with the average of daily energy deficiency (0.623), the total energy deficiency (0.578), the average age (0.585), and the baseline data (BBW: 0.51, BBMI: 0.63, BRMR: 0.68). However, when the decrease in BMI was converted into a percentage decrease, the outcome indicator was only moderately correlated with an average age, BBMI, BRMR, and daily energy deficiency. The reasons may come from the following aspects. First, there is considerable evidence both in the animal model and human beings that confirmed that weight control is more difficult with age [65,66,67,68,69]. Second, many studies confirm that higher BMI is associated with greater bodyweight loss [70,71,72,73,74]. According to Schofield’s RMR formula, higher BMI is associated with higher RMR (the correlation coefficient was 0.876 in correlation analysis). Similarly, in CERG, there was a moderate positive correlation between the average of daily energy deficiency and daily EAT (0.62), which was consistent with the results of the correlation analysis of daily data in the group. Therefore, it could be concluded that the main factors affecting the effect of the CER strategy are age, baseline BMI, and the average daily energy deficiency during the whole plan. Moreover, the key to creating enough daily energy deficiency lies in the daily EAT. Therefore, it could be hypothesized that younger individuals with large bodyweight might achieve a better effect by the CER strategy. At the same time, it is more important to ensure daily energy deficiency than to rigidly restrict energy intake, which means that, for a day, physical activity needs to be increased if the calorie intake is excessive, which is consistent with evidence from previous studies [75,76,77].

After combining the two groups, the correlation analysis of all participants’ data found that the average BMI decrease was only related to that of the daily energy deficiency (0.540) and the total energy deficiency (0.514). After converting the decrease in BMI into a percentage decrease, the correlation with the average daily energy deficiency and that of the total energy deficiency decreased slightly (from 0.540 to 0.466, from 0.514 to 0.438), but the correlation with the daily fat intake increased from −0.438 to −0.534. Therefore, it could be inferred that on the premise of energy restriction, no matter IER or CER strategy is adopted, daily fat intake should be first reduced to ensure to create sufficient energy deficiency. This prediction is consistent with the nutritional recommendations of previous studies [78,79,80].

There are some limitations of this study. Above all, the sample size of the trial was small, which might reduce the statistical power of the data analysis. Studies with larger samples are needed in the future. Second, although Schofield’s Equation has already been demonstrated good validity and reliability, the RMR was determined from not a direct measurement but just calculated according to participants’ bodyweight, making it difficult to represent the metabolic adaptation by the change in RMR, which might merely reflect the change in bodyweight. Future studies should use assessment methods with higher quality to measure RMR. Last, as mentioned in the introduction, for sedentary individuals with normal bodyweight, losing bodyweight in a short term seems more due to their psychological needs. However, this study lacked the measurement of participants’ psychological indicators. It prevents researchers from further analyzing the psychological states of the participants. Future studies should focus more on the measurement from a psychological perspective.

## 5. Conclusions

Conclusion: (1) Both CER and IER are effective energy restriction strategies in the short term. There were only minor adverse effects reported during the study, and no serious adverse events occurred. It can be considered that both strategies are safe in the short term. (2) Both IER and CER could achieve a high degree of completion in a short term under a daily EAT about 300–350 kcal (about 2500 kcal weekly) without any influence on the dietary intervention; (3) No matter undertake CER or IER, it would be important to restrict the daily intake of fat to ensure enough daily energy deficiency; (4) The effect of IER was not sensitive to age and initial bodyweight, but was sensitive to the execution of the dietary plan and the participation of physical exercise, whereas CER does not have high requirements for execution but requires enough daily EAT to create enough daily energy deficiency.

Application advice; (1) IER has a wide range of applications, but high requirements for execution. Limiting daily fat and carbohydrate intake is the key to IER’s effect; (2) The CER may be a better option for younger and heavier individuals. Creating enough total energy deficiency is the key for CER’s effect; (3) This study suggests that for sedentary individuals with normal bodyweight who wants to lose bodyweight quickly in a short term, fat intake should be strictly controlled, and sufficient physical exercise should be guaranteed. In addition, on the premise of sufficient carbohydrate intake, high-quality protein should be given priority.

## Figures and Tables

**Figure 1 ijerph-18-11645-f001:**
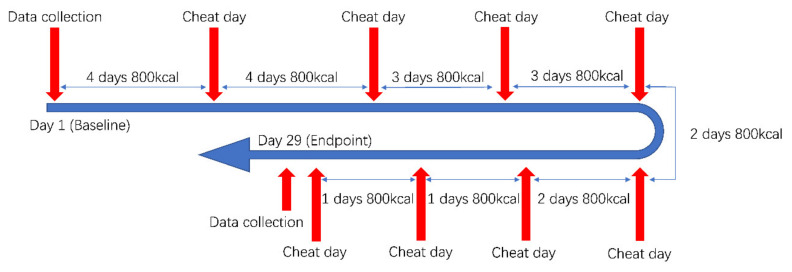
The progress of the intermittent energy restriction intervention.

**Figure 2 ijerph-18-11645-f002:**
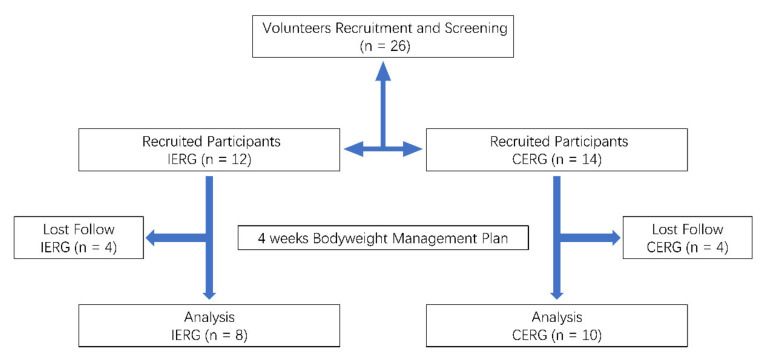
The flow diagram of the whole trial.

**Figure 3 ijerph-18-11645-f003:**
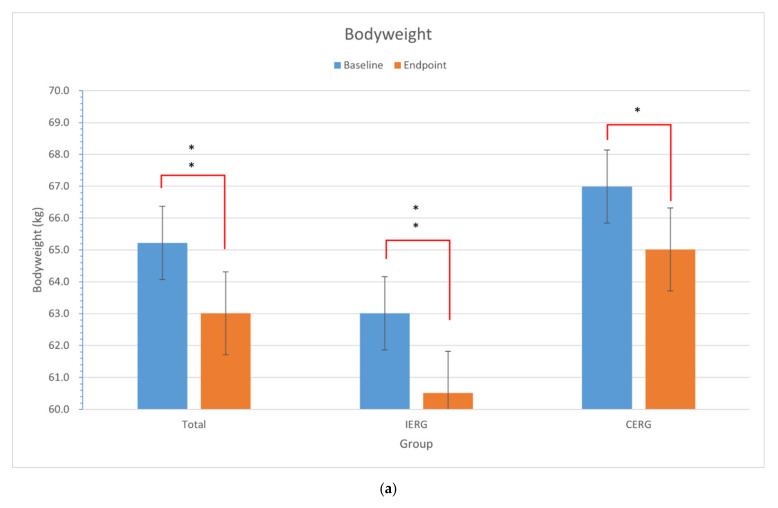
Changes and the difference in bodyweight, BMI, and RMR within and between groups. (**a**) the changes of bodyweight within groups; (**b**) the changes of BMI within groups; (**c**) the changes of RMR within groups. *: *p* < 0.001.

**Table 1 ijerph-18-11645-t001:** Participants’ characteristics in the baseline.

Items	Included after Recruitment				
	Total	IERG	CERG	*t*-Value	Sig
**Recruited (Female)**	26 (22)	12 (11)	14 (11)		
**Analyzed (Female)**	18 (14)	8 (7)	10 (7)		
**Lost rate**	30.8%	33.3%	28.6%		
**Age**	31.3 (6.3)	34.3 (5.1)	29.0 (6.4)	2.358	0.031 *
**BH (cm)**	165.4 (7.8)	163.3 (6.0)	167.1 (8.9)	−2.506	0.023 *
**BW (kg)**	65.2 (11.2)	63.0 (8.5)	67.0 (13.1)	−2.015	0.061
**BMI (kg/m^2^)**	23.8 (3.2)	23.7 (3.2)	23.9 (3.3)	−0.740	0.470
**RMR (kcal)**	1404.0 (230.7)	1340.0 (163.2)	1455.2 (270.6)	−2.018	0.061

Mean (Standard deviation). N: number of participants; BH: body height; BW: bodyweight; BMI: body mass index; RMR: resting metabolic rate. Sig: *p*-value; *: *p* < 0.05 (statistically significant).

**Table 2 ijerph-18-11645-t002:** Results of the paired *t* test of the difference in bodyweight, BMI, and RMR within each group.

Group	BW				BMI				RMR			
	Baseline	Endpoint	*t*	Sig	Baseline	Endpoint	*t*	Sig	Baseline	Endpoint	*t*	Sig
IERG (n = 8)	63.01 (8.46)	60.51 (8.78)	6.859	0.000 **	23.68 (3.19)	22.73 (3.32)	6.971	0.000 **	1339.98 (163.14)	1314.01 (156.79)	4.485	0.003 *
CERG (n = 10)	66.99 (13.09)	65.01 (11.75)	2.294	0.047 *	23.86 (3.28)	23.16 (2.82)	2.384	0.041 *	1455.23 (270.64)	1417.84 (225.67)	1.679	0.128
Total (n = 18)	65.22 (11.15)	63.01 (10.50)	4.446	0.000 **	22.78(3.15)	22.97 (2.97)	4.709	0.000 **	1401.01 (230.67)	1371.69 (199.75)	2.604	0.019 *

Mean (Standard deviation). N: number of participants; BH: body height; BW: bodyweight; BMI: body mass index; RMR: resting metabolic rate. Sig: *p*-value ; *: *p* < 0.05 (statistically significant) ; **: *p* < 0.001.

**Table 3 ijerph-18-11645-t003:** The details of energy balance and the result of the Student’s *t* test between groups with its statistical power.

Items	Group	N	M/MD	SD	Variance Homogeneity		*t* Test			Statistical Power
					F	Sig	Mean Difference	*t*	Sig	1-β
Total Energy Intake (kcal)	IERG	10	28,064	6634	1.113	0.307	−10,664	−3.058	0.008 *	0.78
	CERG	8	38,728	8180						
Total EAT (kcal)	IERG	10	7759	3296	3.993	0.063	−4653	−2.372	0.031 *	0.63
	CERG	8	12,413	5011						
Total Energy Deficiency (kcal)	IERG	10	29,408	12,352	0.072	0.792	−4717	−0.875	0.359	0.78
	CERG	8	34,125	9956						
Change in BW (kg)	IERG	10	2.30	1.03	2.247	0.153	0.188	0.180	0.859	0.92
	CERG	8	2.11	3.10						
Change in BMI (kg/m^2^)	IERG	10	0.88	0.39	2.143	0.163	0.168	0.466	0.647	0.88
	CERG	8	0.71	1.06						
Change in RMR (kcal)	IERG	10	25.95	16.79	3.082	0.098	−14.30	0.561	0.582	0.85
	CERG	8	40.35	78.95						

M/MD: mean/mean difference; SD: standard deviation; Sig: *p*-value; N: number of participants; BH: body height; BW: bodyweight; BMI: body mass index; RMR: resting metabolic rate; EAT: exercise activity thermogenesis; *: *p* < 0.05 (statistically significant).

**Table 4 ijerph-18-11645-t004:** Results of the Student’s *t* test of the sum of valid days between groups and its statistical power.

Items	Group	N	Mean	SD	Variance Homogeneity		*t* test			Statistical Power
					F	Sig	Mean Difference	*t*	Sig	1-β
**VD**	**IERG**	10	26.40	2.72	7.571	0.014 *	−1.475	−1.699	0.122	0.67
	**CERG**	8	27.88	0.35						

VD: valid days; SD: standard deviation; Sig: *p*-value; *: *p* < 0.05 (statistically significant).

**Table 5 ijerph-18-11645-t005:** Correlation analysis between the effect and the variables in the whole plan.

Group	Items	Age	BBH	BBW	BBMI	BMIC	BRMR	R	TCH	TP	TF	TE	TEAT	TED	DCH	DP	DF	DEI	DEAT	DED	PCBMI
**IERG**	**BMIC**	−0.07	−0.06	−0.37	−0.35	1.00	−0.11	0.51 *	0.11	0.31	−0.57 *	−0.48	−0.01	0.26	−0.13	−0.09	−0.72 *	−0.65 *	−0.11	0.20	0.98
	**PCBMI**	0.00	−0.01	−0.47	−0.47	0.98	−0.21	0.53 *	0.12	0.34	−0.59 *	−0.48	−0.02	0.19	−0.12	−0.08	−0.73 *	−0.64 *	−0.12	0.12	1.00
**CERG**	**BMIC**	−0.59 *	0.09	0.51 *	0.63	1.00	0.68 *	−0.05	0.37	−0.11	−0.31	0.00	0.04	0.58 *	0.44	−0.11	−0.36	0.01	0.04	0.62 *	0.98
	**PCBMI**	−0.52 *	0.00	0.38	0.51	0.98	0.55 *	−0.15	0.28	−0.18	−0.37	−0.09	0.03	0.50 *	0.37	−0.16	−0.39	−0.05	0.04	0.55 *	1.00
**Total**	**BMIC**	−0.34	0.01	0.31	0.36	1.00	0.48	0.06	0.12	−0.14	−0.38	−0.14	−0.02	0.51 *	0.11	−0.17	−0.44	−0.17	−0.03	0.54 *	0.97
	**PCBMI**	−0.22	−0.07	0.13	0.19	0.97	0.30	0.02	−0.01	−0.23	−0.48	−0.26	−0.05	0.44	−0.02	−0.25	−0.53 *	−0.29	−0.06	0.47	1.00

BBH: baseline bodyweight; BBMI: baseline body mass index; BMIC: body mass index change; BRMR: baseline resting metabolic rate; R: retention; TCH: total carbohydrate intake; TP: total protein intake; TF: total fat intake; TEAT: total exercise activity thermogenesis; TED: total energy deficiency; DCH: daily carbohydrate intake; DP: daily protein intake; DF: daily fat intake; DEAT: daily exercise activity thermogenesis; DED: daily energy deficiency; PCBMI: percentage change in BMI; *: *p* < 0.05 (statistically significant).

**Table 6 ijerph-18-11645-t006:** Comparison between the mean value of daily statistics between valid and invalid days.

Group	Classification	Subgroups	N	C (g)		*p* (g)		F (g)		DEI (kcal)		EAT (kcal)		DED (kcal)	
				MD	Sig	MD	Sig	MD	Sig	MD	Sig	MD	Sig	MD	Sig
**IERG**	**Energy** **Intake**	**VD**	104	−43.05	0.000 **	−14.91	0.000 **	−19.56	0.000 **	−459.87	0.000 **	−44.22	0.206	345.78	0.000 **
		**IVD**	113												
	**Exercise**	**ED**	167	1.45	0.838	2.07	0.565	1.39	0.665	9.96	0.880	383.39	0.000 **	448.56	0.000 **
		**NED**	50												
**CERG**	**Energy** **Intake**	**VD**	218	−59.54	0.000 **	−17.94	0.010 *	−18.91	0.000 **	−523.14	0.000 **	275.53	0.000 **	865.99	0.000 **
		**IVD**	24												
	**Exercise**	**ED**	195	−8.14	0.303	3.75	0.346	−3.69	0.205	−34.38	0.575	477.37	0.000 **	544.75	0.000 **
		**NED**	47												
**Total**	**Energy** **Intake**	**VD**	332	−10.37	0.060 *	−2.38	0.43	−9.74	0.000 **	−182.77	0.000 **	85.60	0.010 *	307.04	0.000 **
		**IVD**	137												
	**Exercise**	**ED**	362	−0.36	0.950	3.98	0.190	−0.53	0.810	8.92	0.860	434.40	0.000 **	486.39	0.000 **
		**NED**	97												

MD: Mean Difference; Sig: *p*-value; N: number of participants; VD: valid days; IVD: invalid days; ED: exercised day; NED: non-exercised day; C: carbohydrate; P: protein; F: fat; DEI: daily energy intake; EAT: exercise activity thermogenesis; DED: daily energy deficiency; *: *p* < 0.05 (statistically significant); **: *p* < 0.001.

**Table 7 ijerph-18-11645-t007:** Correlation analysis between the effect of the plan and the variables in the daily report.

Group	Items	CH	Protein	Fat	DEI	EAT	DED
**IERG**	**CH**	1.000					
	**Protein**	0.215	1.000				
	**Fat**	0.326	0.451	1.000			-
	**DEI**	0.639 *	0.577 *	0.854 *	1.000		
	**EAT**	0.009	0.087	0.186	0.149	1.000	
	**DED**	−0.482	−0.445	−0.478	−0.677 *	0.506 *	1.000
**CERG**	**C**	1.000					
	**P**	0.283	1.000				
	**F**	0.154	0.440	1.000			-
	**DEI**	0.638 *	0.760 *	0.667 *	1.000		
	**EAT**	0.041	0.290	0.048	0.216	1.000	
	**DED**	−0.341	−0.102	−0.347	−0.357	0.759 *	1.000
**Total**	**CH**	1.000					
	**Protein**	0.283	1.000				
	**Fat**	0.154	0.440	1.000			-
	**DEI**	0.638 *	0.760 *	0.667 *	1.000		
	**EAT**	0.041	0.290	0.048	0.216	1.000	
	**DED**	−0.341	−0.102	−0.347	−0.357	0.759 *	1.000

C: carbohydrate; P: protein; F: fat; DEI: daily energy intake; EAT: exercise activity thermogenesis; DED: daily energy deficiency; *: *p* < 0.05 (statistically significant).

**Table 8 ijerph-18-11645-t008:** The results of regression analysis.

Group	Dependent	Variables	Adjusted R^2^	Constant		Coefficient	
				Constant	Sig	B	Sig
**IERG**	**PCBMI (%)**	**R**	0.157	−11.316	0.310	0.565	0.180
		**TF**	0.234	9.278	0.021 *	−0.005	0.127
		**DF**	0.459	10.319	0.005 *	−0.172	0.039 *
		**DEI**	0.316	11.473	0.020 *	−0.008	0.085
	**DED (kcal)**	**DEI**	0.833	1824.660	0.000 **	−0.982	0.000 **
		**EAT**				1.239	0.000 **
**CERG**	**PCBMI (%)**	**Age**	0.416	3.809	0.664	−0.244	0.163
		**BBMI**				−0.722	0.339
		**BRMR**				0.015	0.183
		**TED**				0.032	0.102
		**DED**				−0.001	0.105
	**DED (kcal)**	**EAT**	0.574	648.899	0.000 **	1.033	0.000 **
**Total**	**PCBMI (%)**	**DF**	0.240	9.660	0.002 *	−0.152	0.022 *
	**DED (kcal)**	**DEI**	0.813	1649.280	0.000 **	−0.794	0.000 **
		**EAT**				1.232	0.000 **

Sig: *p*-value; PCBMI: percentage change of BMI; DED: daily energy deficiency; R: retention; TF: total fat intake; DF: daily fat intake; DEI: daily energy intake; EAT: exercise activity thermogenesis; BBMI: baseline BMI; BRMR: baseline RMR; TED: total energy deficiency. *: *p* < 0.05 (statistically significant); **: *p* < 0.001.

## Data Availability

Not applicable.

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
