# Peer review of "The Comparison of the Effects between Continuous and Intermittent Energy Restriction in Short-Term Bodyweight Loss for Sedentary Population: A Randomized, Double-Blind, Controlled Trial"

_ijerph, 2021, doi:10.3390/ijerph182111645_

Round 1

Reviewer 1 Report

This study investigated the effects between continuous and intermittent energy restriction on body weight loss for sedentary population. The results are partly interesting; however, some serious problems are found as indicated below.

Major comments

  1. I think that energy restriction is important for obese people and athletes as authors described. However, for sedentary people with normal weight, even though I can understand the importance of physical activity, but I can’t understand the necessity of energy restriction and weight reduction for their health.
    This point was partly explained by psychological needs for weight reduction in sedentary people, but it is not supported by objective evidence, and I consider it is not enough to explain the necessity of this study.

  1. Difference of change in RMR between IERG and CERG was discussed. However, since RMR was determined from not direct measurement but participants’ body weight, difference of change in RMR is not means metabolic adaptation as authors discussed, it merely reflects change in body weight.

  1. Authors concluded that there was no difference in weight reduction effect between IERG and CERG. However, because total energy deficiency was not shown, this conclusion is somewhat doubtful. According to the results of average daily energy deficiency (448.6 kcal in IERG and 544.7 kcal in CERG, Discussion) and sum of energy restricted day (26.4 days in IERG and 27.88 days in CERG, Table 4), it can be presumed that total energy deficiency during 4-week program is smaller in IERG than CREG. If this presumption is correct, it provides possibility that IERG is more effective to weight loss because energy deficiency to lose body weight is small.

  1. There is a concern about the reliability of influenced factors which were determined from small sample (IERG; n=8, CERG; n=10). I consider that, at least, statistical reason of sample size and/or study limitation due to small sample size should be described.

Minor comment

  1. L182
    It is necessary to delete “[]”.

  1. Figure 1 and Table 4
    In this study, energy restricted day in IERG was set as 20 days (Figure 1). Why the sum on energy restricted days would be 26.4 days (Table 4)?

  1. Table 3, Table 4
    I consider that comparison between groups was conducted by student’s t-test, not by paired t-test.

  1. Table 5, Table 7
    It would be better to use symbols than use bold characters in order to show significant difference because of legibility.

  1. Table 6
    Item name could be 'DEI' not 'DET', I believe.

  1. Table 7
    There was no explanation in Result section about Table 7. The authors are encouraged to add it.

  1. Table 8
    Unit of the PCBMI could be ‘%’ not 'kg/m2', I believe.

Author Response

Response to Reviewer 1 Comments

Thank you very much for your attention and comments concerning our manuscript entitled “The comparison of the effects between continuous and intermittent energy restriction in short-term bodyweight loss for sedentary population: A randomized, double-blind, controlled trial”. Those comments are all valuable and very helpful for revising and improving our paper, as well as the important guiding significance to our research. We have studied all the comments and suggestions carefully and made corrections, hoping to meet with the final approval. Revised portions are highlighted in red in the revised manuscript. Here below is our description of the revision according to the reviewers’ comments, and all the line numbers correspond to the manuscript displayed in all-visible “Track Changes”.

Major comments

Point 1: I think that energy restriction is important for obese people and athletes as authors described. However, for sedentary people with normal weight, even though I can understand the importance of physical activity, but I can’t understand the necessity of energy restriction and weight reduction for their health.
This point was partly explained by psychological needs for weight reduction in sedentary people, but it is not supported by objective evidence, and I consider it is not enough to explain the necessity of this study.

Response 1: Thank you for your comments. Previous studies have demonstrated that even a small decrease in bodyweight (or BMI) would be beneficial to human health. According to your comments, we added some new cites (Cite No. 26 to 28) and explanations to illustrate the necessity of our trial, as in Line 118-123.

 Point 2: Difference of change in RMR between IERG and CERG was discussed. However, since RMR was determined from not direct measurement but participants’ body weight, difference of change in RMR is not means metabolic adaptation as authors discussed, it merely reflects change in body weight.

Response 2: Thank you for your comments. According to your comments, we added new cites (Cite No. 39 to 41) and explanations to demonstrate that free-living daily energy expenditure adjusted for body composition is relatively constant, as in Line 482 to 486. Moreover, we added the limitation analysis of this issue in the last part of the Discussion section, as in Line 637 to 642.

Point 3: Authors concluded that there was no difference in weight reduction effect between IERG and CERG. However, because total energy deficiency was not shown, this conclusion is somewhat doubtful. According to the results of average daily energy deficiency (448.6 kcal in IERG and 544.7 kcal in CERG, Discussion) and sum of energy restricted day (26.4 days in IERG and 27.88 days in CERG, Table 4), it can be presumed that total energy deficiency during 4-week program is smaller in IERG than CREG. If this presumption is correct, it provides possibility that IERG is more effective to weight loss because energy deficiency to lose body weight is small.

Response 3: Thank you for your comments. According to your suggestion, we added the TED data with its Student’s t-test of participants in IERG and CERG into the Result section, as in Line 345 to 347 and Table 3. Moreover, we added the discussion of the result that there is no statistically significant difference in TED between groups, as in Line 469-475.

Point 4: There is a concern about the reliability of influenced factors which were determined from small sample (IERG; n=8, CERG; n=10). I consider that, at least, statistical reason of sample size and/or study limitation due to small sample size should be described.

Response 4: Thank you for your comments. According to your suggestion, we added an analysis of statistical power in our manuscript, as in Line 303 to 305 in the Methods section, Line 354 to 356, 381 to 382, Table 3, and Table 4 in the Result section, and Line 541 to 544 in the Discussion section. We also added the description of the limitation in the Discussion section, as in Line 635 to 638.

Minor comment

Point 5: L182
It is necessary to delete “[]”.

Response 5: Thank you for your comments. We deleted the “[]”, as in Line 197.

Point 6: Figure 1 and Table 4
In this study, energy restricted day in IERG was set as 20 days (Figure 1). Why the sum on energy restricted days would be 26.4 days (Table 4)?

Response 6: Thank you for your comments. We revised the terms and added their definitions. We replaced the “energy restricted days” and “energy unrestricted days” with the terms “valid days” and “invalid days”, as in Line 203 to 205 and Line 219 to 221.

Point 7: Table 3, Table 4
I consider that comparison between groups was conducted by student’s t-test, not by paired t-test.

Response 7 : Thank you for your comments. We revised them. Sorry for the inconvenience we made.

Point 8: Table 5, Table 7
It would be better to use symbols than use bold characters in order to show significant difference because of legibility.

Response 8: Thank you for your comments. We revised them. Sorry for the inconvenience we made.

Point 9: Table 6
Item name could be 'DEI' not 'DET', I believe.

Response 9: Thank you for your comments. We revised them. Sorry for the inconvenience we made.

Point 10: Table 7
There was no explanation in Result section about Table 7. The authors are encouraged to add it.

Response 10: Thank you for your comments. We added the explanation of Table 7, as in Line 420 to 423.

Point 11: Table 8
Unit of the PCBMI could be ‘%’ not 'kg/m2', I believe.

Response 11: Thank you for your comments. We revised them. Sorry for the inconvenience we made.

Once again, thank you very much for your suggestions and comments, and we feel highly honored by your support!

Reviewer 2 Report

Thank you for the invitation to review this interesting article.

Comparing the effectiveness of two weight reduction programs (CER vs. IER) has a very practical aspect.

The article is well-written and gives a clear picture of the actual state of knowledge.

In my opinion, one of the advantages is the methodology described in great detail and a well-thought-out study protocol.

I have some comments/suggestions.

Line 183 – word ‘per’ is unnecessary

A small group of participants also pay attention and analyzes made on a group of 8/10 individuals.

I have objections about the choice of the scale of an axis. This gives the false impression of spectacular differences

And in my opinion, Figure 4 is not necessary

I have no additional comments

Author Response

Response to Reviewer 2 Comments

Thank you very much for your attention and comments concerning our manuscript entitled “The comparison of the effects between continuous and intermittent energy restriction in short-term bodyweight loss for sedentary population: A randomized, double-blind, controlled trial”. Those comments are all valuable and very helpful for revising and improving our paper, as well as the important guiding significance to our research. We have studied all the comments and suggestions carefully and made corrections, hoping to meet with the final approval. Revised portions are highlighted in red in the revised manuscript. Here below is our description of the revision according to the reviewers’ comments, and all the line numbers correspond to the manuscript displayed in all-visible “Track Changes”.

Point 1: Line 183 – word ‘per’ is unnecessary

Response 1: Thank you for reminding us. We deleted the “per” in the manuscript, as in Line 199.

Point 2: A small group of participants also pay attention and analyzes made on a group of 8/10 individuals.

Response 2: Thank you for your comments. We agreed with the comments and added the study limitation in the discussion section, as in Line 635 to 647.

Point 3: I have objections about the choice of the scale of an axis. This gives the false impression of spectacular differences

Response 3: Thank you for your comments. We revised the scale of the axis.

Point 4: And in my opinion, Figure 4 is not necessary

Response 4: Thank you for your comments. We deleted Figure 4 according to your suggestion.

Once again, thank you very much for your suggestions and comments, and we feel highly honored by your support!

Round 2

Reviewer 1 Report

I thank authors for revising manuscript in accordance with comments; however, serious problem is remained. I believe that following major issues need to be addressed.

Major comments

Authors claimed that “bodyweight loss, even for those who are clinically "normal", is of great value and importance” with some references (Ref 26-28). However, in my understanding, these references do not mention importance of weight loss for normal weight individual.

Therefore, scientific background showing benefits of weight loss for sedentary normal weight individual is still unclear. In addition, even if weight loss is beneficial for them, I cannot comprehend why short-term weight loss is needed.

Authors added limitation as “…, making it difficult to represent the metabolic adaptation by the change in RMR, which might merely reflect the change in body weight.”, as I was suggested. However, authors still discuss about change in RMR and metabolic adaptation (L428-457). These descriptions are potentially confusing.

I can’t agree with claim that change in RMR does mean that a 4-week IER plan might induce a negative metabolic adaption and that change in energy expenditure might not occur in sedentary individuals with general body weight. Again, it is because that presence or absence of significant change in RMR simply depend on amount of change in body weight, in this study.

Author Response

Response to Reviewer 1 Comments

Thank you very much for your attention and new comments concerning our manuscript. Those comments are all valuable and very helpful for revising and improving our paper, as well as the important guiding significance to our research. We have studied all the comments and suggestions carefully and made corrections, hoping to meet with the final approval. Revised portions are highlighted in red in the revised manuscript. Here below is our description of the revision according to the reviewers’ comments, and all the line numbers correspond to the manuscript displayed in all-visible “Track Changes”.

Major comments

Point 1. Authors claimed that “bodyweight loss, even for those who are clinically "normal", is of great value and importance” with some references (Ref 26-28). However, in my understanding, these references do not mention importance of weight loss for normal weight individual.

Therefore, scientific background showing benefits of weight loss for sedentary normal weight individual is still unclear. In addition, even if weight loss is beneficial for them, I cannot comprehend why short-term weight loss is needed.

Response 1. Thank you very much for your new comments. According to your comments, we added some explanation of the study’s value. As in Line 124 to 130, the value of short-term bodyweight loss programs for individuals with normal BMI is from two sides. On one hand, under the precondition that the BMI is in the normal range, there are still health benefits from a decrease in bodyweight. On the other hand, although clinically speaking, individuals with normal bodyweight do not have to lose bodyweight in a short term. However, it is very common to desire to lose bodyweight quickly from the perspective of psychosocial needs and practical application in social life. For example, to attend a wedding, a dinner party, an important interview, etc., in a few weeks. People often encounter similar scenarios and hope to reach their target bodyweight as quickly as possible, to look better and feel better about themselves. This trial aims to estimate the effect of different energy restriction strategies and assess their safety. To enhance our point, we also added a systematic review of cross-sectional and longitudinal observational studies, a qualitative study, and a cross-sectional social study to demonstrate the value of bodyweight loss from a psychosocial perspective, as the Ref. 29-31.

Point 2. Authors added limitation as “…, making it difficult to represent the metabolic adaptation by the change in RMR, which might merely reflect the change in body weight.”, as I was suggested. However, authors still discuss about change in RMR and metabolic adaptation (L428-457). These descriptions are potentially confusing.

I can’t agree with claim that change in RMR does mean that a 4-week IER plan might induce a negative metabolic adaption and that change in energy expenditure might not occur in sedentary individuals with general body weight. Again, it is because that presence or absence of significant change in RMR simply depend on amount of change in body weight, in this study.

Response 2. Thank you very much for your new comments. According to your suggestion, we rewrote the discussion of the RMR. Firstly, we added the explanation of the preconditions for RMR calculation and analysis, as in Line 483 to 490. The equation for calculating RMR in this study is based on the body composition characteristics of a large sample population. Since the BMI of the participants in this study was within the normal range, we assumed that the body composition of the participants was consistent with the body composition characteristics of the population applicable to this equation. Second, we discussed the difference in metabolic adaption between IERG and CERG could not be proved. We listed our analysis from three perspectives, which were population differences, intervention protocol differences, and durational differences, as in Line 505 to 526. Lastly, we suggested that future studies should explore what is the best energy restriction strategy for a rapid bodyweight loss plan in sedentary people with normal bodyweight by tracking the change of their body composition and metabolic hormone response, as in Line 527 and 530. Moreover, we revised the conclusion about safety, removed the part about RMR, and merely stated that there were only minor adverse effects reported during the study, and no serious adverse events occurred, as in Line 671 to 673.
